# SPEAR: STRUCTURED PRUNING FOR SPIKING NEURAL NETWORKS VIA SYNAPTIC OPERATION ESTIMATION AND REINFORCEMENT LEARNING

## ABSTRACT

While deep spiking neural networks (SNNs) demonstrate superior performance, their deployment on resource-constrained neuromorphic hardware still remains challenging. Network pruning offers a viable solution by reducing both parameters and synaptic operations (SynOps) to facilitate the edge deployment of SNNs, among which search-based pruning methods search for the SNNs structure after pruning. However, existing search-based methods fail to directly use SynOps as the constraint because it will dynamically change in the searching process, resulting in the final searched network violating the expected SynOps target. In this paper, we introduce a novel SNN pruning framework called SPEAR, which leverages reinforcement learning (RL) technique to directly use SynOps as the searching constraint. To avoid the violation of SynOps requirements, we first propose a SynOps prediction mechanism called LRE to accurately predict the final SynOps after search. Observing SynOps cannot be explicitly calculated and added to constrain the action in RL, we propose a novel reward called TAR to stabilize the searching. Extensive experiments show that our SPEAR framework can effectively compress SNN under specific SynOps constraint.

## 1 INTRODUCTION

Recently, Spiking Neural Networks (SNNs) have attract many attention because of their high energy efficiency and superior performance (Zhou et al., 2024; Luo et al., 2025; Su et al., 2024; Lei et al., 2024; Qiu et al., 2024a). However, the limited resources of edge neuromorphic hardware (Bouvier et al., 2019; Pei et al., 2019; Davies et al., 2018) hinder the deployment of deep SNNs. Structured pruning technology can effectively compress SNN to reduce the network parameters and computation, making it a viable solution for the deployment SNNs on edge neuromorphic devices.

Current SNN structured pruning methods can be categorized into design-based methods (Di Yu et al., 2024; Chowdhury et al., 2021) and search-based methods (Li et al., 2024a;b). Design-based methods prune the channel based on channel importance criteria. As these approaches need to manually design the pruned network structure, the network architecture after pruning remains sub-optimal. Search-based approaches utilize network architecture search technique to automatically search for the optimal pruned network under a specific constraint. However, as one of the most important metric for energy efficiency (Shi et al., 2024; Yan et al., 2024), existing methods fail to directly use synaptic operations (SynOps) as the constraint in the searching process. As a result, the SynOps remains high after compression by using existing pruning approaches (see Table. 1), making the deployment of compressed model challenging.

It is a non-trivial task to use SynOps as the constraint for searching. First, different from the FLOPs in artificial neural networks (ANNs), SynOps will significantly change after finetuning, which is a standard operation in the pruning approaches. Fig. 1(a) shows the SynOps of pruned models before and after finetuning. We observe SynOps exhibit irregular and significant variations after the finetuning operation. Therefore, the searched network may violate or deviate far beyond the constraints after finetuning. On the other hand, if we use SynOps after finetuning as the constraint, the time-consuming training procedure will make the search unaffordable. Therefore, it is desirable

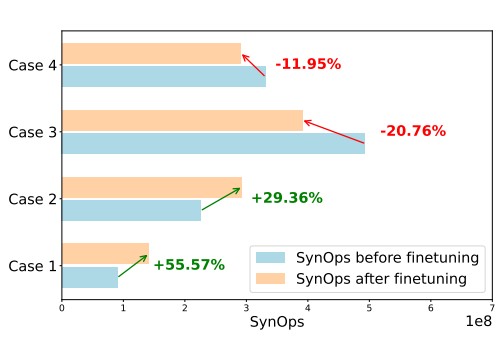

(a) SynOps vary before and after finetuning

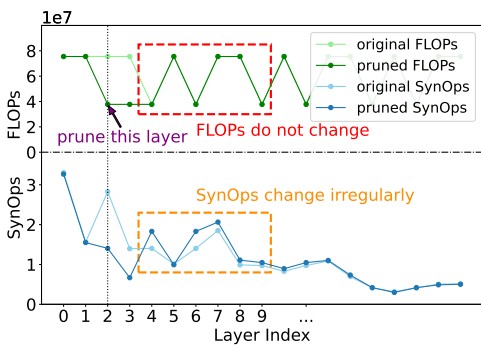

(b) Pruning effect layer-wise SynOps and FLOPs

Figure 1: Characteristics of SynOps in finetuning and pruning

to design a mechanism to accurately and efficiently estimate the SynOps after finetuning for effective search.

Second, reinforcement learning has been proved as effective method to search for pruned network structure. Current reinforcement learning based ANN pruning methods (He et al., 2018; Ganjdanesh et al., 2024) utilize the property that FLOPs can be explicitly calculated based on the given formula. So, they use FLOPs as the explicit constraint for the action at each step. For example, AMC (He et al., 2018) uses remained FLOPs budget to limit the sparsity ratio when pruning each layer. However, SNNs do not hold this property. SynOps in SNN is related to many factors, which cannot be directly calculated based on the formula. As shown in Fig. 1(b), the SynOps of other layers will also change when we prune one layer in SNN, which makes it challenging to calculate the remained SynOps budget to limit the action when pruning. As a result, it is also desirable to incorporate SynOps constraint implicitly at each action step for efficient searching.

To address the aforementioned issues, in this paper, we propose an SNN pruning framework called SPEAR, which leverages reinforcement learning to automatically compress the network. To accurately estimate post-finetuning SynOps, we first propose linear regression for SynOps Estimation (LRE) strategy to predict the final SynOps based on pre-finetuning SynOps. Specifically, observing the linear correlation between pre- and post-finetuning SynOps, we propose to use linear regression to directly learn the relationship of SynOps at different stages, which provides effective and efficient SynOps estimation.

To tackle the second problem (i.e., use SynOps budget to limit action), we further propose a novel reward function called target-aware reward (TAR). Specifically, instead of using hard SynOps constraint in the searching process, we seamlessly integrate SynOps penalty in our reward function and penalize the reward when the current SynOps exceeds the target constraint. By converting hard constraint to soft penalty, we can smoothly optimize our RL agent to meet the resource limitation.

Our contributions are summarized as follows:

- We propose the SPEAR structured pruning framework, which leverages reinforcement learning to automatically compress SNNs.
- We reveal that SynOps will irregularly and significantly change after finetuning, and propose LRE strategy to accurately predict the final SynOps by using linear regression.
- We design a novel reward function called TAR, which can smoothly optimize the RL agent and enforce it to meet the resource constraint.
- Extensive experiments on various datasets demonstrate our SPEAR method can effectively compress SNN.

## 2 RELATED WORK

**Spiking Neural Networks.** Recently, spiking neural networks have attracted attention because of their superior performance and energy efficiency. Many network architectures (Sengupta et al.,

2018; Hu et al., 2021; Fang et al., 2021; Yao et al., 2024; 2025; Qiu et al., 2024b; Yao et al., 2021; 2023) were proposed. For example, Fang et al. (2021) proposed SEW ResNet to overcome the vanishing/exploding gradient problems of Spiking ResNet. In addition to the architecture design, other approaches like design different neurons (Hao et al., 2023; Yao et al., 2022; Huang et al., 2024), or improve training techniques (Guo et al., 2022; 2023; Deng et al., 2022; Zheng et al., 2021) were also proposed to improve the SNN performance. However, these methods aim to either improve SNNs performance or improve the training efficiency. In contrast, our SPEAR framework aims to compress these SNN for efficient deployment, which is complementary to these approaches.

**Neural Network Pruning.** Neural network pruning has garnered growing attention for building efficient deep learning systems by removing redundant parameters. ANN pruning (He & Xiao, 2024; Ghimire et al., 2022; Liu et al., 2021; Guo et al., 2020) has been well explored in recent years. For example, AMC (He et al., 2018) uses reinforcement learning to automatically search the compressed network structure. Ganjdanesh et al. (2024) proposed to use reinforcement learning to dynamically learn the weights and architecture. Compared with ANN pruning approaches, our SPEAR aims to compress SNNs, where the dynamically changed SynOps is the main metric for efficiency measurement. So, we propose LRE and TAR to effectively estimate the final SynOps and to integrate constraint into the searching process.

Recently, there are also SNN pruning approaches in the literature (Han et al., 2025; Chen et al., 2021; 2022; Shi et al., 2024; Li et al., 2024a; Liu et al., 2017; Garg et al., 2019; Chen et al., 2023). For example, Chowdhury et al. (2021) proposed to use principal component analysis on membrane potential to determine the channel width. However, these SNN pruning approaches require manual design of the network structure after pruning, which is sub-optimal. There are also searching-based pruning approaches to automatically search the pruned network structure. Li et al. (2024b) proposed to prune and regenerate convolutional kernels based on their activity levels. These methods cannot effectively decrease the SynOps after compression as they do not directly use SynOps as the constraint in the searching process. Although Shi et al. (2024) utilizes SynOps as the constraint for pruning neurons and weights, this approach focuses on unstructured pruning, which is hardware unfriendly. In contrast, our SPEAR focuses on structured pruning, which is hardware-friendly and can achieve practical acceleration.

**Neural Architecture Search for SNN.** Neural Architecture Search (NAS) (Yan et al., 2024; Liu et al., 2024) aims to automatically design and optimize SNN architectures through searching algorithms. For example, SNASNet (Kim et al., 2022) uses temporal feedback connections for searching. SpikeDHS (Che et al., 2022) adapts the Darts (Liu et al., 2018) to search for the surrogate gradient function. AutoSNN (Na et al., 2022) takes both the accuracy and energy efficiency into account and uses one-shot architecture search paradigm. However, these approaches focus on searching for the operations or connections types. In contrast, our SPEAR framework aims to search the channel width of each layer to compress an existing SNN, which is complementary to these methods.

## 3 PRELIMINARY

### 3.1 SYNAPTIC OPERATIONS

The primary metric for evaluating the energy consumption of neuromorphic chips is the average energy required to transmit a single spike through a synapse (Furber, 2016). Therefore, the number of synaptic operations is an important metric to measure the efficiency of a model. Following previous work (Shi et al., 2024), we define the number of synaptic operations as follows:

$$\text{SynOps} = \sum_k s_k \times c_k, \tag{1}$$

where SynOps denotes the total number of synaptic operations for one sample. $s_k$ and $c_k$ denote the number of spikes fired by neuron $k$ and the number of synaptic connections from neuron $k$, respectively. As the SynOps for each sample can be different, so we define the average SynOps on the dataset as follows:

$$\text{Avg. SynOps} = \frac{\sum_j^N \text{SynOps}_j}{N}, \tag{2}$$

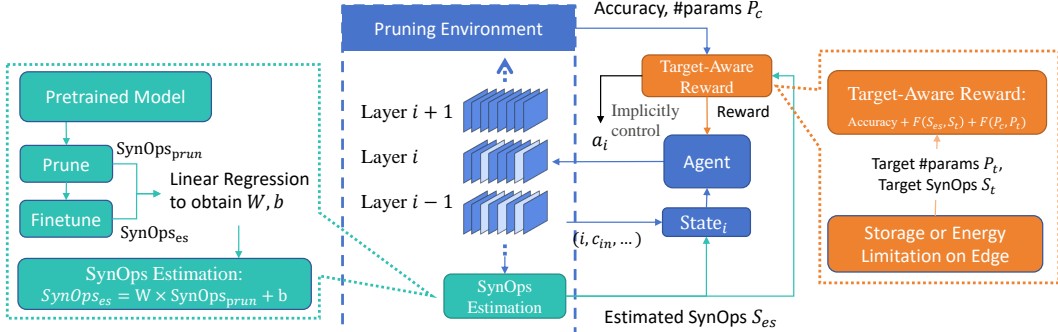

Figure 2: Overview of our SPEAR framework.

where $\text{SynOps}_j$ means the SynOps of the $j$-th sample, and $N$ denotes number of samples in the dataset. For simplicity, the term SynOps mentioned in the following paper refers to the average SynOps, unless otherwise specified.

## 4 METHODOLOGY

### 4.1 OVERVIEW

The overview of our Structured Pruning for SNNs via Synaptic Operation Estimation and Reinforcement Learning (SPEAR) framework is shown in Figure 2. We aim train a reinforcement learning agent based on environmental feedback to achieve optimal accuracy under given resource constraints. To provide realistic state feedback, we utilize our proposed Linear Regression for SynOps Estimation (LRE) to predict post-finetuning SynOps based on pre-finetuning SynOps. In the training process, we also introduce the Target-Aware Reward (TAR) to effectively incorporate the SynOps constraint into each action.

### 4.2 LINEAR REGRESSION FOR SYNOPS ESTIMATION

**Motivation.** As introduce in Sec.1, the SynOps of pruned models change significantly after finetuning. If we directly use pre-finetuning SynOps as the constraint in the searching process, the final network may violate the SynOps constraint after finetuning. Conversely, if we use post-finetuning SynOps as the constraint, the time-consuming finetuning process will make the searching process intractable. Therefore, we propose our linear regression SynOps estimation (LRE) strategy to predict the SynOps after finetuning.

**Linear Regression for SynOps Estimation.** To precisely predict the final SynOps, we first visualize the SynOps before and after finetuning to look for the relationship between them. Specifically, we randomly generate the pruning ratio for each layer (i.e., pruning policy) and use the L1-norm criterion(Li et al., 2017a) to prune the pre-trained SNN based on the generated policy. Then, we calculate the SynOps before and after finetuning of these pruned networks and plot them in Fig. 3. Surprisingly, we observe a linear correlation between the SynOps of pruned models and their finetuned counterparts. This observation enables predictive modeling of post-finetuning SynOps through simple linear regression.

Mathematically, the estimated SynOps after finetuning can be written as follows:

$$\text{SynOps}_{\text{es}} = W \cdot \text{SynOps}_{\text{cur}} + b, \tag{3}$$

where $\text{SynOps}_{\text{es}}$ and $\text{SynOps}_{\text{cur}}$ are the estimated SynOps after finetuning and the actual SynOps before finetuning, respectively. $W$ and $b$ are the learnable parameters. We can sample a small number of pruning policy and prune and finetune them to generate SynOps data. Then, we perform linear regression to fit the SynOps before and after finetuning and obtain learned $W$ and $b$. After learning the LRE, we directly use the learned weight $W$ and bias $b$ to predict the post-finetuning SynOps as Eq. (3).

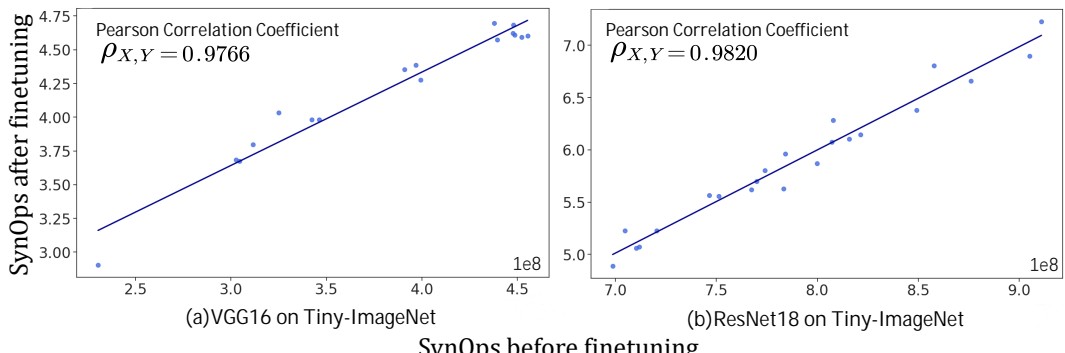

Figure 3: Linear relationship of SynOps pre and post finetuning

### 4.3 TARGET-AWARE REWARD

**Motivation.** In existing reinforcement learning based pruning work (He et al., 2018; Ganjdanesh et al., 2024), constraints on computation were explicitly imposed by bounding action ranges based on the formulaically computed FLOPs of current layer, i.e., the computed FLOPs is used as a constraint for the agent when take action. However, as introduced in Sec 1, SynOps of other layers will also change when pruning a specific layer in SNN because of altered spiking activity patterns. This makes it challenging to use SynOps for limiting actions of the agent.

**Target-Aware Reward.** To this end, we propose Target-Aware Reward (TAR) to implicitly bound the pruning actions of agent by incorporating resource information into the reward during reinforcement learning training. Mathematically, TAR can be written as follows:

$$R_s = Acc + F(S_{es}, S_t),$$

$$\text{where } F(S_{es}, S_t) = -\lambda \cdot \left[ \max\left( \frac{S_{es}}{S_t} - 1, 0 \right) \right]^{\alpha}. \tag{4}$$

Here, $S_{es}$, $S_t$ denotes estimated and target SynOps respectively. $Acc$ means the accuracy of pruned model on validation dataset. The $\max(\cdot, 0)$ operator creates unilateral penalty, ensuring punishment only triggers when current SynOps exceeds the target. The exponential term $[\cdot]^{\alpha}$ with $\alpha > 1$ establishes exponential penalty growth, where excessive SynOps incurs rapidly escalating costs. $\lambda$ is the coefficient to balance accuracy rewards and constraint enforcement intensity. We empirically set $\alpha = 1.2$ and $\lambda = 1$.

In addition to incorporating SynOps into the reward, we can also implicitly use number of parameters or both SynOps and parameters into our TAR. Specifically, the reward using parameters as the constraint can be written as follows:

$$R_p = Acc + F(P_c, P_t), \tag{5}$$

and the reward using both SynOps and parameters is:

$$R_{sp} = Acc + F(S_{es}, S_t) + F(P_c, P_t). \tag{6}$$

$P_c$, $P_t$ denote current and target number of parameters respectively. By using either Eq. (5) or Eq. (6), we can penalize either parameters or both SynOps and parameters in our searching algorithm to achieve the target. For SynOps-dominated scenarios, we can use only $R_s$ as our reward. On the other hand, for parameter-dominated scenarios, we can use only $R_p$. More commonly and practically, $R_{sp}$ is used when both computation and storage resources are limited. By converting the hard constraint to soft penalty in our reward function, we can gradually push the pruning policy close to the target, which provides smoother optimization trajectories.

### 4.4 REINFORCEMENT LEARNING SEARCH

In our SPEAR, we use the TAR as the reward and employ deep deterministic policy gradient (DDPG) (Lillicrap, 2015) to search the pruning policy.

---

**Algorithm 1:** Our SPEAR framework

---

**Input:** Pretrained SNN $M_{pre}$; Validation dataset $D_{val}$; Reinforcement learning agent $\pi$; Total
      episode $num\_episode$; Warmup episode $warmup\_episode$.

\# Linear Regression for SynOps Estimation
Randomly generate pruning policies, then prune and finetune $M_{pre}$ to generate SynOps data for
  LRE;
Use obtained SynOps data and Eq. (3) to learn SynOps estimator $\mathcal{E}$, which can predict
  post-finetuning SynOps;

\# Agent Training
Initialize $episode = 1$;
**while** $episode \leq num\_episode$ **do**
    **for** $i = 0$ *to* $L$ **do**
        **if** $episode \leq warmup\_episode$ **then**
            Sample action $a_i$ from truncated normal distribution;
        **else**
            Calculate pre-finetuning SynOps and estimate final SynOps based on estimator $\mathcal{E}$;
            Obtain the state $State_i$ for the $i$-th layer;
            Predict action $a_i$ using agent $\pi$ based on $State_i$;
        Prune this layer based on $a_i$;
    Evaluate current pruned model on $D_{val}$ and calculate reward using target-aware reward;
    Push trajectory to replay buffer and update agent $\pi$;
    $episode = episode + 1$

\# Apply agent to prune model and finetune
Prune the model $M_{pre}$ using agent $\pi$;
Fine tune the pruned model and generate compressed model $M_{com}$.
**Output:** Compressed model $M_{com}$

---

**State Space.** We first define the state for the agent. Specifically, when pruning the $i$-th layer, the state $State_i$ is described by following features:

$$State_i = (i, c_{\text{in}}, c_{\text{out}}, s, k, p, S_{\text{es}}, P_{\text{cur}}, P_{\text{rest}}, a_{i-1}). \tag{7}$$

Here, $i$ is the index of the layer. $c_{\text{in}}$ and $c_{\text{out}}$ are the number of input and output channels for the $i$-th layer, respectively. $s$, $k$, and $p$ are the stride, the kernel size, and the number of parameters of this layer, where $p = c_{\text{out}} \times c_{\text{in}} \times k \times k$. $S_{\text{es}}$ is the remained SynOps, which is obtained by using Eq. (3) in our LRE for better estimation. $P_{\text{cur}}$ and $P_{\text{rest}}$ are the current number of parameters and the remained parameters that can be removed in subsequent layers, respectively. $a_{i-1}$ is the action taken in the previous layer $i - 1$. All the features in the state are normalized to [0,1] by dividing by the maximum value. This formulation provides a comprehensive description of the state of each layer, capturing both its structural properties and dynamic behavior during the pruning.

**Agent.** We use DDPG for pruning on continuous space. For action space, we use a continuous action space of [0, 1) as the pruning ratio for each layer. Specifically, for the $i$-th layer, our agent takes $State_i$ as the input and output action $a_i$ for this layer. During exploration, we employ a truncated normal distribution with standard deviation of 0.5 for the action sampling. During exploitation, the agent incorporates noise into its actions, sampled from a truncated normal distribution with an initial standard deviation of 0.5. This standard deviation decays exponentially at a rate of 0.98 per episode. Discount factor for reward is set to 1, and only the reward from last action is calculated through our TAR, while the rest are set to 0. Algorithm 1 shows the process of our SPEAR framework.

## 5 EXPERIMENTS

### 5.1 EXPERIMENTAL SETTINGS

**Datasets and Models.** To verify the effectiveness of the proposed method, we carried out experiments on both static datasets and neuromorphic datasets. For static datasets, we conduct experiments on CI-FAR10, CIFAR100 (Krizhevsky et al., 2009), Tiny-ImageNet (Le & Yang, 2015) and ImageNet (Deng

Table 1: Performance comparison between our SPEAR and baseline methods. "-" indicates results are not reported in original paper. "∗" means our implementation.

| Dataset | Arch. | Method | SynOps(%) | Param.(%) | Top-1 Acc.(%) |
|---------|-------|--------|-----------|-----------|---------------|
| **CIFAR10** | VGG16 | NetworkSliming (Li et al., 2024a) | 87.3 | 40.3 | 91.22 |
| | | SCA-based (Li et al., 2024b) | 67.8 | 28.4 | 91.67 |
| | | **SPEAR (Ours)** | **52.5** | **14.4** | **91.77** |
| | ResNet18 | NetworkSliming (Li et al., 2024a) | - | 30.9 | 92.31 |
| | | SCA-based (Li et al., 2024b) | 88.0 | 40.6 | 92.48 |
| | | **SPEAR (Ours)** | **39.2** | **30.3** | **92.78** |
| **CIFAR100** | VGG16 | NetworkSliming (Li et al., 2024a) | - | 40.9 | 66.36 |
| | | SCA-based (Li et al., 2024b) | 82.6 | 42.5 | 66.88 |
| | | **SPEAR (Ours)** | **69.0** | **35.0** | **70.50** |
| | | NetworkSliming (Li et al., 2024a) | - | 20.2 | 63.44 |
| | | SCA-based (Li et al., 2024b) | 77.9 | 23.5 | 65.53 |
| | | **SPEAR (Ours)** | **48.2** | **20.4** | **68.86** |
| **Tiny-ImageNet** | VGG16 | SCA-based (Li et al., 2024b) | - | 43.2 | 49.36 |
| | | **SPEAR (Ours)** | **69.5** | **39.0** | **59.47** |
| | | SCA-based (Li et al., 2024b) | - | 30.6 | 49.14 |
| | | **SPEAR (Ours)** | **37.8** | **23.3** | **56.62** |
| **ImageNet** | ResNet18 | SCA-based∗ (Li et al., 2024b) | 85.8 | 68.4 | 60.17 |
| | | **SPEAR (Ours)** | **84.6** | **65.1** | **60.69** |
| | | SCA-based∗ (Li et al., 2024b) | 76.1 | 40.7 | 59.44 |
| | | **SPEAR (Ours)** | **73.3** | **57.2** | **60.00** |
| **CIFAR10-DVS** | 5Conv+1FC | SCA-based (Li et al., 2024b) | 56.9 | 21.7 | 73.70 |
| | | **SPEAR (Ours)** | **39.3** | **17.1** | **80.05** |

et al., 2009). We copy the images 4 times along the timeline to obtain input for 4 time steps. On static datasets, we adopt ResNet18 (He et al., 2016) and VGG16 (Simonyan & Zisserman, 2015) for evaluation. For neuromorphic datasets, we use CIFAR10-DVS (Li et al., 2017b) for evaluation, in which 8,000 samples are used as training set, and 2,000 samples are used as test set. For each sample, we evenly split the original event stream data into 10 segments, integrating over each segment to obtain input for 10 time steps. We use the same network as in (Li et al., 2024b) called 5Conv+1FC for fair comparison.

**Implementation details.** We use SpikingJelly (Fang et al., 2023) to implement our SPEAR framework. Specifically, we first compress the pre-trained SNN using our SPEAR approach. We use $R_{sp}$ in our TAR by default. Specifically, we use the ratio of SynOps and #parameters over those from pre-trained model as the target. We adjust the target SynOps and #parameters to achieve similar SynOps/#parameters as the baseline methods for fair comparison. For VGG16, we set the target ratio ranging from 0.4 to 0.8 for SynOps and 0.2 to 0.4 for #parameters, respectively. For ResNet18, we set the target ratio ranging from 0.4 to 0.8 for SynOps and 0.3 to 0.6 for #parameters, respectively. For 5Conv+1FC, we set the target ratio ranging from 0.4 to 0.8 for SynOps and 0.2 to 0.5 for #parameters, respectively. After compression, we finetune the compressed SNN for 210 epochs in the same configuration as training to recover from accuracy drop. More experimental details can be found in the Appendix A.1.

## 5.2 EXPERIMENTAL RESULTS

**Results on static datasets.** We show the experimental results on static datasets in Table 1. We have the following observations:

(1) Compared to other baseline methods, our method can achieve higher SynOps compression rates because we directly use estimated post-finetuning SynOps as the constraint, which effectively avoid the SynOps constraint violation after finetuning. For instance, we achieve 60.8% SynOps reduction with higher accuracy on CIFAR10 when compressing ResNet18 compared to other approaches.

(2) Our SPEAR framework maintains higher accuracy at larger compression rates when compared to other approaches. For example, we achieve 91.77% Top-1 accuracy using 14.4% #parameters and 52.5% SynOps, surpassing SCA-based (Li et al., 2024b) approach (91.67% accuracy with 67.8% SynOps and 28.4% #parameters) and NetworkSliming (Li et al., 2024a) (91.22% accuracy with 87.3% SynOps and 40.3% #parameters) when compress VGG16 on CIFAR10.

(3) Our SPEAR framework also outperforms other baseline methods on various datasets when compressing different network architectures, which further demonstrates the effectiveness of our SPEAR. For instance, the compressed VGG16 using our SPEAR achieves 59.47% Top-1 accuracy with 39.0% #parameters on Tiny-ImageNet, which is 10.11% higher than SCA-based (Li et al., 2024b) (49.36% Top-1 accuracy) with similar number of parameters. Also, our SPEAR can outperform other baseline methods on large-scale dataset ImageNet with less SynOps and parameters.

**Results on neuromorphic datasets.** In Table 1, we also report experimental results on neuromorphic dataset CIFAR10-DVS. Our SPEAR framework outperforms the baseline method SCA-based by a large margin under similar SynOps, which further demonstrate the effectiveness of our SPEAR framework on neuromorphic datasets. Other observations are similar to static datasets. So we do not provide further analysis here. In the Appendix A.2, additional experimental results are provided for further reference.

## 5.3 ABLATION STUDIES

To validate the effectiveness of each component in SPEAR framework, we take compressing VGG16 on CIFAR10 as an example and conduct extensive ablation studies.

Table 2: Ablation study for SPEAR on CIFAR10.

| Method | SynOps (%) | #Param. (%) | Acc. (%) |
|---|---|---|---|
| SPEAR | 46.37 | 11.86 | 91.62 |
| SPEAR (w/o LRE) | 61.13 | 19.13 | 91.95 |
| SPEAR (w/o TAR) | 43.21 | 11.33 | 91.14 |

Table 3: Comparison with handcraft pruning policy.

| Pruning policy | Acc. (%) | SynOps (%) | #Param. (%) |
|---|---|---|---|
| Handcrafted | 91.64 | 63.9 | 33.9 |
| Ours | 92.49 | 62.5 | 33.1 |

Table 4: Sensitive analysis for hyper-parameter in TAR[4].

| $\lambda$ | $\alpha$ | Acc. (%) | #Params (%) | SynOps (%) |
|---|---|---|---|---|
| 1.0 | 0.8 | 91.36 | 19.50 | 52.09 |
| 1.0 | 1.0 | 91.43 | 24.69 | 52.01 |
| 1.0 | 1.2 | 91.51 | 26.40 | 52.85 |
| 1.0 | 1.6 | 91.73 | 29.90 | 55.25 |
| 1.0 | 2.0 | 91.70 | 26.23 | 56.06 |
| 0.1 | 1.2 | 92.24 | 30.19 | 73.18 |
| 0.5 | 1.2 | 91.49 | 20.42 | 52.91 |
| 1.0 | 1.2 | 91.51 | 26.40 | 52.85 |
| 5.0 | 1.2 | 91.76 | 30.57 | 53.12 |
| 10.0 | 1.2 | 90.91 | 29.66 | 49.38 |

**Effect of LRE.** To demonstrate the effectiveness of LRE, we directly use the pre-finetuning SynOps as the constraint and set the SynOps constraint as 50% for search. The result is denoted as "SPEAR (w/o LRE)" in Table 2. We observe that the final pruned network exceeds 50% SynOps constrains after fine-tuning, showing that it is important to estimate post-finetuning SynOps as the target in the searching process.

**Effect of TAR.** To validate the effectiveness of TAR, we substitute our TAR with the reward in RL-pruner(Wang & Kindratenko, 2024), which is a widely used reinforcement learning based pruning approach, and the result is denoted as "SPEAR (w/o TAR)" in Table 2. We observe our SPEAR surpasses this alternative method by 0.48% under similar SynOps, which demonstrates the effectiveness of our TAR.

**Comparison with handcrafted pruning policy.** For handcrafted pruning method, we follow AMC (He et al., 2018) to assign lower pruning rate to shallow layers and higher rate for deep layers. As shown in Table 3, our SPEAR can outperforms handcrafted methods.

**Sensitive analysis of TAR.** We conduct experiments with various $\lambda$ and $\alpha$ values, and the results are reported in Table 4. Results show that our SPEAR is robust to different values of $\lambda$ and $\alpha$. Note $\lambda$ controls the penalty strength for SynOps violation. If it is set too low (e.g., 0.1), our SPEAR may can not reduce SynOps significantly as the reward for accuracy outweighs the SynOps penalty. If it is set too high (e.g., 10), our SPEAR may fail to fully explore the search space, resulting in a sub-optimal performance.

**Comparison between LRE and nonlinear SynOps estimator.** We also compare our LRE with nonlinear regression for SynOps estimation as shown in Fig. 4. For nonlinear estimator, we use a two-layer MLP for SynOps estimation. MSE represents the root mean square error (smaller is better) between the prediction and the actual SynOps, and $R^2$ is the coefficient of determination (Rider,

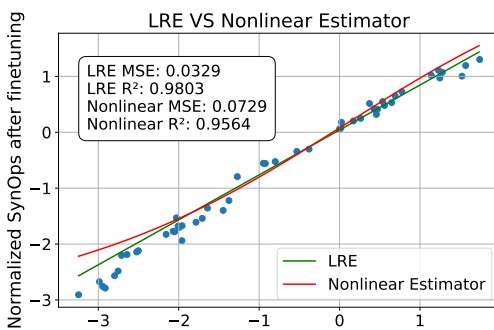

Figure 4: Comparison between linear regression and nonlinear regression.

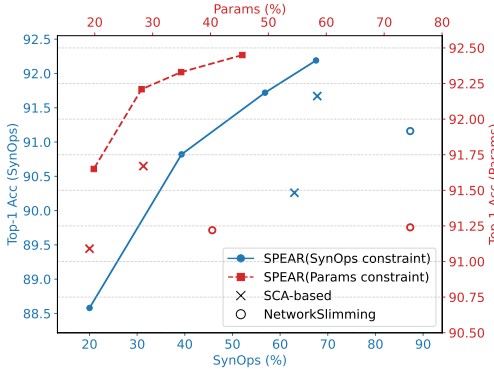

Figure 5: Results under different compression ratios with different constrains.

1932) (higher is better). We observe that our linear LRE is better than nonlinear regression with smaller root mean square error and higher $R^2$.

**Analysis on different target constraints.** To demonstrate the generalization ability of our target-aware reward, we also conduct the experiments when using $R_s$ and $R_p$ under different target ratios, and the results are shown in Fig. 5. From Fig 5, we can surpass other baseline methods under different compression ratios when using both SynOps and #parameter constraints, which demonstrate that our SPEAR can generalize to different penalty types under different scenarios.

Table 5: Energy consumption and speedup comparison.

| Model | Acc. (%) | #Add. (M) | #Mult. (M) | Energy (mJ) | Latency (%) | Speedup | SynOps (%) | #Param. (%) |
|---|---|---|---|---|---|---|---|---|
| VGG16 (ANN) | 93.36 | 626.4 | 626.4 | 2.88 | - | - | - | 100 |
| VGG16 (SNN) | 92.43 | 107.6 | 3.54 | 0.11 | 100 | 1× | 100 | 100 |
| Ours | 92.49 | 68.1 | 3.1 | 0.07 | 59.9 | 1.67× | 62.5 | 33.1 |

**Energy consumption and speedup analysis of SPEAR.** In Table 5, we follow (Horowitz, 2014; Yan et al., 2024; Che et al., 2022; Yin et al., 2024) to report the energy consumption and latency of different approaches on 45nm CMOS chip. We observe VGG16 (SNN) model can significantly reduce the energy cost compared to its counterpart VGG16 (ANN). Moreover, our compressed model can further reduce the energy cost compared to VGG16 (SNN) and also achieve 1.67× speedup, with higher performance. The results demonstrate our SPEAR can achieve practical energy saving and speedup.

Table 6: Algorithm Efficiency Comparison

| Method | Time (hours) |
|---|---|
| NetworkSliming | 2.1 |
| SCA-based | 2.8 |
| Ours | 2.4 |

**Efficiency analysis of SPEAR.** To demonstrate the efficiency of our SPEAR framework, we also report the training time for compression as Table 6. Our SPEAR only requires 2 hours to complete the search, which shows the proposed method is efficient to compress pretrained SNNs for edge deployment.

## 6 CONCLUSION

In this paper, we proposed Structured Pruning for Spiking Neural Networks via Synaptic Operation Estimation and Reinforcement Learning (SPEAR), which employs reinforcement learning algorithms to automatically explore optimal network architectures under specific SynOps and parameter constraints. To the best of our knowledge, our SPEAR is the first SynOps-oriented structured pruning framework. We reveal that SynOps will change irregularly and significantly after fine-tuning. Therefore, we propose Linear Regression for SynOps Estimation (LRE) strategy to accurately predict post-finetuning SynOps based on pre-finetuning SynOps. Additionally, we also propose a novel Target-Aware Reward (TAR) function that adapt to search under various constraint scenarios, enabling implicit control on the action space of agent through reward. Experiments on various datasets demonstrate the effectiveness of our SPEAR framework.

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

# A APPENDIX

## A.1 MORE IMPLEMENTATION DETAILS

Our experiments are based on Leaky Integrate-and-Fire (LIF) neurons with a hard reset mechanism. We set the fire threshold as 1.0, and set membrane potential time constant as 2.0. No decay for input currents is used. We use arctan function as the surrogate function. For pretrained model preparation, we use the SGD optimizer with momentum of 0.9 for optimization. Weight decay is set as $5 \times 10^{-5}$. We train 210 epochs to obtain the pretrained models. In the first 10 epochs, we employ linear warm-up strategy. The following 200 epochs adopt cosine annealing schedule with 0.1 as max learning rate. TET (Deng et al., 2022) is used as loss function. For static datasets, no data augmentation is applied, while for neuromorphic datasets, random erasing is utilized. All experiments are conducted on an NVIDIA A800 GPU.

**Pruning strategy.** After the agent predicts the pruning ratio for each layer, channels are pruned based on the L1-norm criterion Li et al. (2017a), where channels with the smallest L1-norm of weights are removed. This process is repeated in a layer-by-layer fashion, and the pruned network is then finetuned to recover accuracy. We find using L1-norm criterion for channel selection is effective and efficient. We will explain the pruning strategy in detail in our final version.

**Efficient SynOps Calculation.** As SynOps is a statistical measurement calculated over the dataset, it is computationally prohibitive to iterate through the entire dataset for each evaluation in Eq. (3). To address this challenge, we design an accelerated sampling strategy to use a small number of samples to estimate dataset-level SynOps. Specifically, to evaluate the estimation error, we first define the SynOps relative error as:

$$Error = \frac{\text{Sample SynOps} - \text{Dataset SynOps}}{\text{Dataset SynOps}}, \tag{8}$$

where "Sample SynOps" and "Dataset SynOps" denote the average SynOps computed over sampled subsets and the entire dataset, respectively. We iteratively sample more data from the dataset, and compute the error in Eq. (8). When the error is smaller than a predefined value, we use the current subset to learn the parameters in Eq. (3), which serves as reliable and computationally efficient proxy for full-dataset computation. In our implementation, we empirically set the error tolerance as 1% and find 500 samples are sufficient.

In our LRE strategy, we sample a small subset as the proxy of datasets SynOps. We iteratively sample more data from the dataset until the error in Eq. (8) is smaller than a predefined value. Then, we use the current subset to learn the parameters in Eq. (3), which serves as reliable and computationally efficient proxy for full-dataset computation. In our implementation, we empirically set the error tolerance as 1% and find 500 samples are sufficient.

## A.2 MORE PERFORMANCE RESULTS

In this section, we present comparative experimental results under various configurations to validate the effectiveness of our approach in Table 7. Additionally, we report the experimental outcomes of ResNet18 on the CIFAR100 and Tiny-ImageNet dataset in Table 8. We further include the performance and SynOps of both the pretrained model and the pretrained model undergoes the same finetune process as the pruned model (finetuned model) in Table 9. We can see that the performance and SynOps of the pretrained model and finetuned model are similar. Also, our SPEAR framework can effectively reduce SynOps and parameters while preserving the model accuracy. Note that SynOps will change after finetuning the pretrained model, which is widely recognized in SNN area.

Table 7: Detailed performance comparison between our SPEAR and baseline methods. "-" indicates results are not reported in original paper. "∗" means our implementation.

| Dataset | Arch. | Method | SynOps(%) | Param.(%) | Top-1 Acc.(%) |
|---|---|---|---|---|---|
| CIFAR10 | VGG16 | NetworkSliming (Li et al., 2024a) | 87.3 | 40.3 | 91.22 |
| | | SCA-based (Li et al., 2024b) | 67.8 | 28.4 | 91.67 |
| | | **SPEAR (Ours)** | **62.5** | **33.1** | **92.49** |
| | | NetworkSliming (Li et al., 2024a) | 87.3 | 14.3 | 91.16 |
| | | **SPEAR (Ours)** | **52.5** | **14.4** | **91.77** |
| | | SCA-based (Li et al., 2024b) | 63.0 | 9.3 | 90.26 |
| | | **SPEAR (Ours)** | **46.4** | **11.9** | **91.62** |
| | ResNet18 | NetworkSliming (Li et al., 2024a) | - | 48.2 | 92.71 |
| | | **SPEAR (Ours)** | **71.5** | **50.1** | **93.99** |
| | | SCA-based (Li et al., 2024b) | 88.0 | 40.6 | 92.48 |
| | | **SPEAR (Ours)** | **56.6** | **35.9** | **93.71** |
| | | NetworkSliming (Li et al., 2024a) | - | 30.9 | 92.31 |
| | | SCA-based (Li et al., 2024b) | 84.0 | 27.8 | 92.27 |
| | | **SPEAR (Ours)** | **39.2** | **30.3** | **92.78** |
| CIFAR100 | VGG16 | NetworkSliming (Li et al., 2024a) | - | 40.9 | 66.36 |
| | | SCA-based (Li et al., 2024b) | 82.6 | 42.5 | 66.88 |
| | | **SPEAR (Ours)** | **69.0** | **35.0** | **70.50** |
| | | NetworkSliming (Li et al., 2024a) | - | 20.2 | 63.44 |
| | | SCA-based (Li et al., 2024b) | 77.9 | 23.5 | 65.53 |
| | | **SPEAR (Ours)** | **48.2** | **20.4** | **68.86** |
| Tiny-ImageNet | VGG16 | SCA-based (Li et al., 2024b) | - | 43.2 | 49.36 |
| | | **SPEAR (Ours)** | **69.5** | **39.0** | **59.47** |
| | | SCA-based (Li et al., 2024b) | - | 30.6 | 49.14 |
| | | **SPEAR (Ours)** | **61.2** | **32.1** | **58.84** |
| | | **SPEAR (Ours)** | **37.8** | **23.3** | **56.62** |
| CIFAR10-DVS | 5Conv+1FC | **SPEAR (Ours)** | **77.6** | **50.8** | **82.30** |
| | | **SPEAR (Ours)** | **47.2** | **40.0** | **81.80** |
| | | SCA-based (Li et al., 2024b) | 56.9 | 21.7 | 73.7 |
| | | **SPEAR (Ours)** | **39.3** | **17.1** | **80.05** |
| | | SCA-based (Li et al., 2024b) | 39.5 | 7.0 | 71.9 |
| | | **SPEAR (Ours)** | **33.0** | **11.4** | **79.75** |

Table 8: Performance of SPEAR for ResNet18 on CIFAR100 and Tiny-ImageNet.

| Dataset | Arch. | Method | SynOps(%) | Param.(%) | Top-1 Acc.(%) |
|---|---|---|---|---|---|
| CIFAR100 | ResNet18 | **SPEAR (Ours)** | **81.2** | **50.1** | **75.08** |
| | | **SPEAR (Ours)** | **59.7** | **38.9** | **74.58** |
| | | **SPEAR (Ours)** | **42.3** | **33.7** | **73.25** |
| Tiny-ImageNet | ResNet18 | **SPEAR (Ours)** | **80.0** | **62.4** | **62.28** |
| | | **SPEAR (Ours)** | **64.3** | **50.1** | **61.30** |
| | | **SPEAR (Ours)** | **44.6** | **39.8** | **60.37** |

## A.3 MORE ANALYSIS

**Analysis on Pruning Policy.** To investigate the detailed structures of pruned network, in Fig. 6, we also visualize the searched pruning policy and the SynOps distribution across different layers. We observe deeper layers often have higher pruning ratios, which indicates the deeper layers are less sensitive to the pruning. Moreover, for SynOps target ($R_s$) pruning, the pruning ratio for shallow layers are higher when compared to the parameter target ($R_p$) pruning. We hypothesize this is because shallow layers consists of more SynOps compared to deeper layers. Therefore, it is beneficial to prune shallow layers in this case. On the other hand, for parameter target pruning, the pruning ratio is almost zero for shallow layers, which shows that the parameters may be more important in these layers compared to deep layers. Furthermore, the pruning ratio obtained by $R_{sp}$ for shallow layers are between SynOps and parameter targets policies, which shows our SPEAR can effectively adjust the pruning policy to meet both constraints in the searching process.

**Comparison of Reinforcement Learning Algorithms.** To evaluate the sensitivity of our method to the choice of reinforcement learning algorithm, we additionally conducted experiments using the

Table 9: Comparison of SPEAR with pretraining and finetuning baselines.

| Dataset | Arch. | Method | SynOps ($\times 10^8$) | Param.(%) | Top-1 Acc.(%) |
|---|---|---|---|---|---|
| CIFAR10 | VGG16 | **SPEAR (Ours)** | **0.650** | **33.1** | **92.49** |
| | | **Finetune** | 0.935 | 100.0 | 92.50 |
| | | **Pretrain** | 1.04 | 100.0 | 92.43 |
| CIFAR10 | ResNet18 | **SPEAR (Ours)** | **1.61** | **50.1** | **93.99** |
| | | **Finetune** | 2.17 | 100.0 | 94.47 |
| | | **Pretrain** | 2.25 | 100.0 | 94.18 |
| CIFAR100 | VGG16 | **SPEAR (Ours)** | **0.86** | **35.0** | **70.50** |
| | | **Finetune** | 1.19 | 100.0 | 70.62 |
| | | **Pretrain** | 1.24 | 100.0 | 70.75 |
| Tiny-ImageNet | VGG16 | **SPEAR (Ours)** | **3.48** | **39.0** | **59.47** |
| | | **Finetune** | 4.80 | 100.0 | 59.98 |
| | | **Pretrain** | 5.00 | 100.0 | 60.50 |
| Tiny-ImageNet | ResNet18 | **SPEAR (Ours)** | **8.48** | **50.1** | **62.28** |
| | | **Finetune** | 10.3 | 100.0 | 63.17 |
| | | **Pretrain** | 10.6 | 100.0 | 63.10 |
| ImageNet | ResNet18 | **SPEAR (Ours)** | **8.58** | **65.1** | **60.69** |
| | | **Finetune** | 10.18 | 100.0 | 62.21 |
| | | **Pretrain** | 10.14 | 100.0 | 62.08 |
| CIFAR10-DVS | 5Conv+1FC | **SPEAR (Ours)** | **0.463** | **50.8** | **82.30** |
| | | **Finetune** | 0.612 | 100.0 | 83.60 |
| | | **Pretrain** | 0.595 | 100.0 | 84.05 |

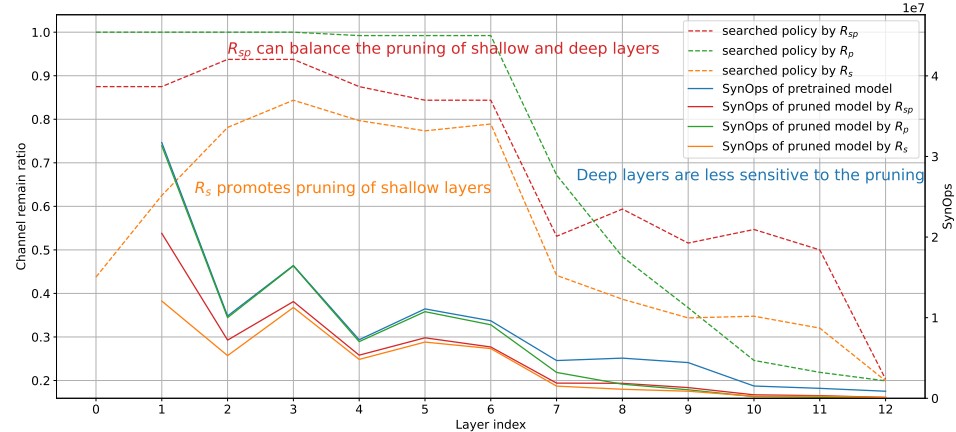

Figure 6: The pruning policy and SynOps distribution of each layer under different targets given by our reinforcement learning agent.

Soft Actor-Critic (SAC) algorithm Haarnoja et al. (2018), a widely adopted off-policy RL method in continuous control tasks. The comparative results, summarized in Table 10, demonstrate that both DDPG and SAC achieve comparable performance.

Table 10: Performance comparison of DDPG and SAC in SPEAR under different compression levels.

| Method | Top-1 Acc.(%) | Param.(%) | SynOps(%) |
|---|---|---|---|
| Pretrained | 92.43 | 100.0 | 100.0 |
| SAC | 92.39 | 33.4 | 60.4 |
| DDPG | 92.49 | 33.1 | 62.5 |
| SAC | 91.69 | 17.2 | 49.5 |
| DDPG | 91.62 | 11.9 | 46.4 |

Table 11: Empirical validation of soft constraint adherence under various target compression ratios. Target Type" denotes the resource metric being constrained (Synaptic Operations or Parameters). Target Ratio" is the specified compression goal, Remain Ratio" is the actual ratio achieved by the searched strategy.

| Target Type | Target Ratio (%) | Remain Ratio (%) | Top-1 Acc. (%) |
|---|---|---|---|
| SynOps | 20 | 19.8 | 88.64 |
| | 40 | 39.4 | 90.69 |
| | 60 | 57.3 | 91.84 |
| | 70 | 66.4 | 92.26 |
| Params | 20 | 19.6 | 91.67 |
| | 30 | 28.4 | 92.21 |
| | 40 | 35.2 | 92.37 |
| | 50 | 44.6 | 92.46 |

**Empirical Validation of Soft Constraint Satisfaction.** While our formulation employs soft constraints to guide the architecture search, empirical evidence indicates that the resulting strategies reliably satisfy the prescribed resource limits. As shown in Table 11, across a range of stringent compression targets for both synaptic operations (SynOps) and model parameters, demonstrating that the obtained strategies consistently meet the prescribed resource budgets with high fidelity, even under stringent compression targets.

**Robustness of SPEAR to Varying Timesteps.** To address concerns regarding the influence of timestep count on SynOps estimation, pruning strategy, and overall performance, we evaluate our SPEAR method across different timesteps on CIFAR10 using a VGG-16 architecture. As shown in Table 12, SPEAR consistently achieves competitive accuracy and SynOps reduction across timesteps 2, 4, and 6. We also validate the LRE under different time steps, and find that LRE is robust to different timesteps (linear regression MSE: 0.0049 and coefficient of determination $r^2$: 0.9859 for timesteps 2, linear regression MSE: 0.0060 and $r^2$: 0.9864 for timesteps 6).

Table 12: Robustness of SPEAR to varying timesteps on CIFAR10 with VGG-16.

| Time Step | Pretrained Acc (%) | Pretrained SynOps ($\times 10^8$) | Pruned Acc (%) | Pruned SynOps (%) | Pruned Params (%) |
|---|---|---|---|---|---|
| 2 | 89.70 | 0.43 | 89.75 | 67.0 | 31.3 |
| 4 | 92.43 | 1.04 | 92.49 | 62.5 | 33.1 |
| 6 | 92.41 | 1.39 | 92.54 | 71.2 | 30.0 |

**Training Convergence under Target-Aware Reward.** To evaluate the learning dynamics and stability of the proposed Target-Aware Reward mechanism, we monitored the average reward throughout training. Empirical results show that, when using TAR as the reward signal, the agent's average reward converge rapidly (around 200 steps) to a stable and optimal value as training progresses, confirming the effectiveness of our TAR formulation and the stability of the learned policy. Training curves illustrating this convergence behavior are included in Figure 7.

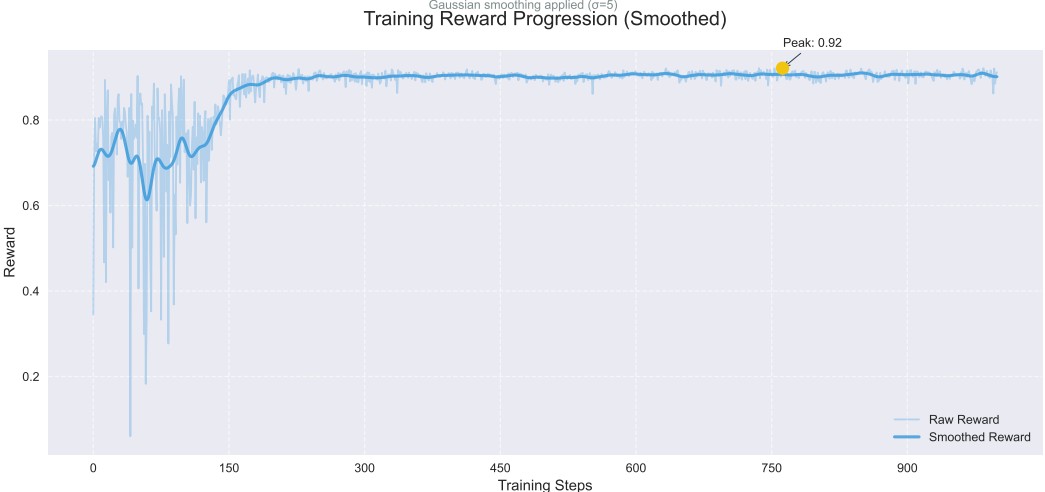

Figure 7: Training curves of average reward over episodes for the proposed Target-Aware Reward mechanism.

