# OpenReview forum: "SPEAR: Structured Pruning for Spiking Neural Networks via Synaptic Operation Estimation and Reinforcement Learning"
_ICLR.cc/2026/Conference — ICLR 2026 Conference Withdrawn Submission_

### Official Review · Reviewer_gRyG · 2025-10-26

**Soundness:** 2
**Presentation:** 2
**Contribution:** 3
**Rating:** 4
**Confidence:** 4

**Summary:**

This paper presents SPEAR, a reinforcement learning-based structured pruning framework for SNNs, which directly uses SynOps as a constraint. It demonstrates that a simple linear regression (LRE) can be used to predict post-finetuning SynOps. Innovatively, it combines LRE with reinforcement learning, enforcing resource constraints implicitly during the search process through a Target-Aware Reward (TAR). Experimental results across multiple datasets demonstrate that SPEAR achieves superior compression rates compared to existing methods.

**Strengths:**

1.**Well-motivated problem**. The observation that SynOps change significantly and irregularly after finetuning is important for SNN deployment and distinguishes this work from ANN pruning methods.

2.**Practical and effective approach**. The LRE method, despite its simplicity, achieves strong performance with low computational overhead. The TAR design cleverly converts hard constraints to soft penalties, enabling smoother optimization.

**Weaknesses:**

1. **Limited theoretical foundation of LRE**: While a linear relationship is empirically observed, the paper does not investigate the underlying mechanism or the conditions under which it applies. Furthermore, using 500 samples to learn two parameters (W and b) appears inefficient and requires further clarification.

2. **Limited experimental comparison**: Although SPEAR primarily addresses search-based structured pruning, a more comprehensive comparison with design-based, as well as other search-based structured and unstructured pruning methods, would be valuable .

**Questions:**

1.A comprehensive comparison with related works should be included, covering  and search-based methods that use SynOps for unstructured pruning [1]，and  also based on  SCA-based design approaches [2], with a focus on SynOps, parameter count, and performance.Additionally, comparisons with SNN architecture search works, as mentioned in the related work section, should be incorporated for a more thorough evaluation.

2.Provide a theoretical analysis or intuition for why the relationship between pre- and post-finetuning SynOps is approximately linear.

3.How often must the LRE surrogate be retrained (e.g., for new target SynOps, new datasets, or new architectures)? What is the marginal cost compared to overall SPEAR training time?

[1] "Towards energy efficient spiking neural networks: An unstructured pruning framework." ICLR. 2024.
[2] “Qp-snn: Quantized and pruned spiking neural networks." ,ICLR. 2025.

---

### Official Review · Reviewer_ZFV1 · 2025-10-27

**Soundness:** 3
**Presentation:** 3
**Contribution:** 2
**Rating:** 4
**Confidence:** 3

**Summary:**

The paper introduces SPEAR, a SynOps-constrained structured pruning framework for SNNs. It proposes LRE, a linear regression model that accurately estimates post-finetuning SynOps, and TAR, a reinforcement learning reward function that smoothly enforces resource constraints. Extensive experiments show that SPEAR achieves higher accuracy, better compression, and greater energy efficiency across multiple benchmarks.

**Strengths:**

Originality
1. The paper integrates SynOps constraints into structured SNN pruning using reinforcement learning.
2. The TAR reward formulation elegantly transforms hard resource constraints into soft penalties, enabling smooth optimization.

Quality
1. The paper demonstrates technical rigor, with detailed methodology, theoretical motivation, and algorithmic clarity.
2. Extensive quantitative results across both static and neuromorphic datasets validate generalizability.
3. Ablation studies systematically isolate the effects of LRE and TAR, supporting the claimed contributions.

Clarity
The writing is clear and well-structured, figures and tables effectively illustrate problem motivation, algorithm design, and empirical results.

Significance
Addresses a key bottleneck for deploying deep SNNs on edge devices. The integration of SynOps-aware pruning can influence future SNN model compression and neuromorphic design studies.

**Weaknesses:**

1. Limited novelty in RL formulation: While the paper integrates reinforcement learning into SNN pruning, the use of DDPG and reward shaping is largely inspired by existing ANN pruning frameworks. The novelty lies mainly in the application to SynOps constraints rather than a fundamentally new RL algorithm.
2. Comparison limited to few baselines: The evaluation primarily compares against NetworkSlimming and SCA-based pruning, which provides a limited perspective. Including more recent and diverse SNN pruning or NAS methods would strengthen the experimental validation and make the results more convincing.

**Questions:**

1. Generalization of LRE: Does the linear correlation between pre- and post-finetuning SynOps hold for all network architectures (e.g., SNN-Transformers or temporal attention-based models)? Could a nonlinear or adaptive estimator further improve accuracy?
2. The comparison currently focuses on NetworkSlimming and SCA-based pruning. Could the authors include more diverse baselines to strengthen the empirical analysis?

---

### Official Review · Reviewer_8nC4 · 2025-10-28

**Soundness:** 3
**Presentation:** 3
**Contribution:** 3
**Rating:** 4
**Confidence:** 4

**Summary:**

This paper proposes SPEAR, a structured pruning framework SNNs that leverages reinforcement learning and introduces Synaptic Operation (SynOps) estimation to directly control energy efficiency during compression. SPEAR directly targets SynOps, a biologically meaningful and hardware-relevant measure of energy use in SNNs. Linear Regression for SynOps Estimation predicts post-finetuning SynOps from pre-finetuning SynOps via a linear regression model, which avoids costly retraining. Target-Aware Reward ensures the reinforcement learning agent learning compression policies that meet resource targets without violating constraints.

**Strengths:**

1. SPEAR’s focus on SynOps aligns closely with energy efficiency on neuromorphic hardware, which is relevant to SNN training.
2. The combination of linear regression estimation and RL-based optimization is elegant and practical and demonstrates that linear estimation is sufficient and more stable than nonlinear approaches.
3. This paper is well-structured and clear written, with detailed explanation of each component and motivation.

**Weaknesses:**

1. These is a green rectangular on page 6 and 7. This should be corrected in the final version.
2. The linear relationship assumption between pre- and post-finetuning SynOps is empirically validated but lacks formal theoretical grounding.
3. Though not excessive, cost-benefit trade-offs for extremely large SNNs are unexplored.

**Questions:**

1. How well does linear regression for SynOps estimation generalize when the pruning ratios or datasets differ significantly from those used to train the regression model?
2. Would a nonlinear SynOps estimator (e.g., shallow MLP) improve estimation accuracy on larger datasets or more complex networks?
3. How stable is the reinforcement learning search: does it require many episodes to converge, and how sensitive is it to reward scaling?
4. Are there any more timesteps settings using for this study? Though timestep=4 is a small timestep setting, would there be any chance for a less timestep?

---

### Official Review · Reviewer_QTBN · 2025-11-02

**Soundness:** 2
**Presentation:** 2
**Contribution:** 2
**Rating:** 4
**Confidence:** 3

**Summary:**

This work proposes a structured pruning algorithm for SNNs. The method is based on the principle of neural architecture search, incorporating the number of synaptic operations as one of the key constraints. To enable faster estimation of the number of synapses, the authors introduce a linear regression–based estimation method. In addition, a reward function is designed for the reinforcement learning process to encourage architectures with an appropriate #SynOps.

**Strengths:**

+ The paper is clearly written and well structured, making it easy to follow the methodology and contributions.

+ The work introduces SynOps as a constraint within the NAS process, which is straightforward.

**Weaknesses:**

- NAS encompasses various search paradigms, including RL-based, gradient-based, and evolutionary approaches. Why do the authors specifically adopt an RL-based framework for deriving the pruning strategy? A more detailed justification and comparison with alternative NAS strategies would strengthen the methodological motivation.

- Although the paper empirically shows a linear correlation between pre-fine-tuning and post-fine-tuning SynOps, the proposed estimation technique lacks theoretical grounding and analysis of generalizability. Could the authors further clarify the rationale behind the linear regression model? Additionally, more discussion is needed on why the reward formulation in Equations (4), (5), and (6) is theoretically sound.

- From a hardware perspective, the energy evaluation remains simplified. Despite frequent references to “hardware,” the work does not validate results on an actual neuromorphic platform such as TrueNorth [1] or Loihi [2]. Incorporating real hardware experiments would significantly improve the paper’s credibility regarding energy efficiency claims. More importantly, measuring SNN execution on physical devices may reveal that SynOps is not always the dominant contributor to energy consumption, which would challenge a core assumption of the proposed method.

[1]Akopyan, Filipp, et al. "Truenorth: Design and tool flow of a 65 mw 1 million neuron programmable neurosynaptic chip." IEEE transactions on computer-aided design of integrated circuits and systems 34.10 (2015): 1537-1557.
[2]Davies, Mike, et al. "Loihi: A neuromorphic manycore processor with on-chip learning." Ieee Micro 38.1 (2018): 82-99.

**Questions:**

See the weakness.

---

### Note · Authors · 2025-11-14

I have read and agree with the venue's withdrawal policy on behalf of myself and my co-authors.